# ViSER: Video-Specific Surface Embeddings for Articulated 3D Shape Reconstruction

**Gengshan Yang**[1]    **Deqing Sun**[2]    **Varun Jampani**[2]    **Daniel Vlasic**[2]    **Forrester Cole**[2]

**Ce Liu**[4*]    **Deva Ramanan**[1,3]

[1]Carnegie Mellon University    [2]Google Research    [3]Argo AI    [4]Microsoft Azure AI

`ViSER-webpage`

## Abstract

We introduce ViSER, a method for recovering articulated 3D shapes and dense 3D trajectories from monocular videos. Previous work on high-quality reconstruction of dynamic 3D shapes typically relies on multiple synchronized cameras, strong category-specific priors, or 2D keypoint supervision. We show that none of these are required if one can reliably estimate long-range correspondences in a video, making use of only 2D object masks and two-frame optical flow as inputs. ViSER infers correspondences by matching 2D pixels to a canonical, deformable 3D mesh via *video-specific surface embeddings* that capture the view-independent appearance features of each surface point. These embeddings behave as a continuous set of keypoint descriptors defined over the mesh surface, which can be used to establish dense long-range correspondences across pixels. The surface embeddings are implemented as coordinate-based MLPs that are fit to each video via self-supervised losses. Experimental results show that ViSER compares favorably against prior work on challenging videos of humans with loose clothing and unusual poses as well as animal videos from DAVIS and YTVOS.

## 1 Introduction

Reconstructing the world from a sequence of monocular frames is a long-standing task in computer vision. While there has been tremendous progress in reconstructing rigid scenes (via SfM and SLAM [7, 39, 43], or recent techniques based on neural rendering [28]), reconstructing *dynamic* scenes with articulated objects remains elusive. For example, given a monocular video, it is still challenging to reconstruct an everyday scene of a moving person with loose clothing. In this work, we tackle the problem of estimating the deforming mesh of articulated objects given a segmented monocular video of that object. Our method avoids the use of any mesh templates or category-specific priors and generalizes to unknown deformable articulated objects in the wild.

Nonrigid shape recovery is highly under-constrained due to fundamental ambiguities between shape, appearance, and time-varying deformation. Current approaches for addressing these challenges fall into two camps: better data "likelihoods" or better "priors". The first camp extracts richer sensor data, via multi-camera studio setups [15] or depth sensors [30], but requires substantial efforts to work in the wild. The second camp makes use of category-level priors over object shapes [18, 20] and is particularly effective for human reconstruction. However, building such models requires considerable offline efforts in the form of registered 3D scans [26] or manual keypoint annotations [12], both of which are difficult to scale to arbitrary object categories.

---

*Work done at Google.

35th Conference on Neural Information Processing Systems (NeurIPS 2021).

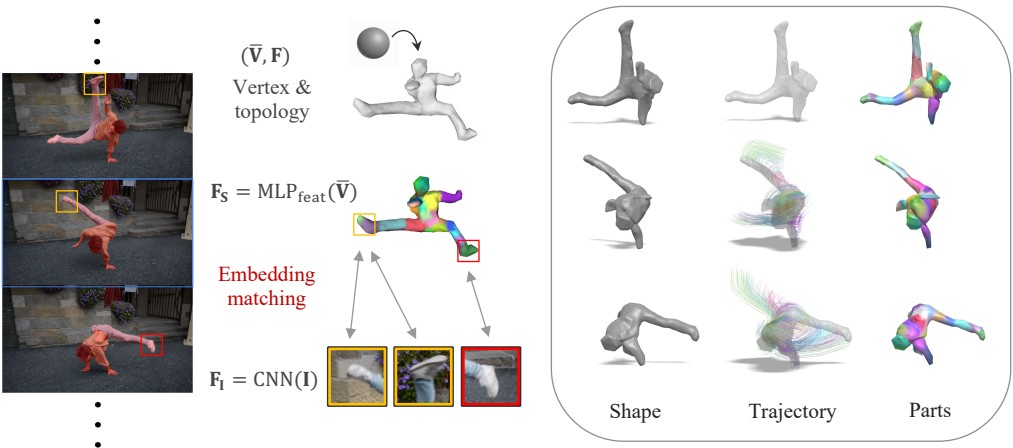

Figure 1: Given a long video (or multiple short videos), ViSER jointly learns articulated 3D shapes (represented as a mesh with vertices $\bar{\mathbf{V}}$ and faces $\mathbf{F}$) and joint pixel-surface embeddings (including a surface embedding $\mathbf{F_S}$ and a pixel embedding $\mathbf{F_I}$) that establishes dense long-range pixel correspondences over time. As a result, ViSER produces accurate shapes, long term trajectories and meaningful part segmentation.

In this work, we use a practical but less explored variant of the data-likelihood camp: we use *multiple frames* of a video rather than multiple cameras or depth sensors. This considerably complicates analysis for dynamic, non-rigid scenes. Nonrigid structure-from-motion (NRSfM) [4, 38] attempts to constrain the problem by relying on motion correspondences such as 2D point tracks. While 2D correspondences over short time scales (i.e., optical flow) are relatively robust to extract, correspondences over long time scales are notoriously difficult to estimate because of appearance variations arising from viewpoint changes, occlusion and fast motion. In practice, this limits the applicability of NRSfM methods to controlled lab sequences.

We propose ViSER (Video-Specific Surface Embeddings for Reconstruction), which establishes long-range correspondence and reconstructs articulated 3D shapes from a monocular video. Fig. 1 shows a sample outdoor video and the corresponding ViSER results. The key insight behind ViSER is to force long-range video pixel correspondences to be consistent with an underlying canonical 3D mesh through the use of video-specific embeddings that capture the pixel appearance of each surface point. These embeddings behave as a continuous set of keypoint descriptors defined over the surface mesh, learned with coordinate-based MLPs that are fit to each video via self-supervised losses. ViSER simultaneously optimizes the image CNN, surface MLP, and 3D shape so as to fit the observed video frames. It reconstructs state-of-the-art articulated 3D shape and 3D trajectories without using category-specific priors, making it easily scalable to diverse videos including humans with challenging clothing and poses as well as animals. Lastly, we demonstrate that ViSER recovers meaningful part segmentation and blend skinning weights from videos, which typically require considerable manual effort from 3D artists.

## 2   Related Work

**Low-level correspondence.** Optical flow is a well-studied representation for short-term correspondence between adjacent frames of a video. After decades of research, recent CNN models [40, 42, 48] for optical flow have achieved an impressive level of accuracy as evidenced by the Sintel and KITTI benchmarks [5, 9]. However, it is challenging to concatenate optical flow for reliable long-range correspondence due to occlusions and strong appearance changes [33, 37, 41]. ViSER does not concatenate optical flow but use it as a constraint to establishes long-range correspondence.

The layered approach [6, 14, 45] segments a video into different moving objects with coherent motion, thereby establishing long-range correspondence for every frames through the shared layers. Early layered methods assume parameter motion for each layer and can only handle limited scenes. Unwrap Mosiacs [32] uses a dense 2D-to-2D mapping from a texture map to every input frame, and editing operations on the texture map naturally transfers to each individual frame. However, the 2D representation cannot flexibly model complex 3D phenomena, such as occlusions.

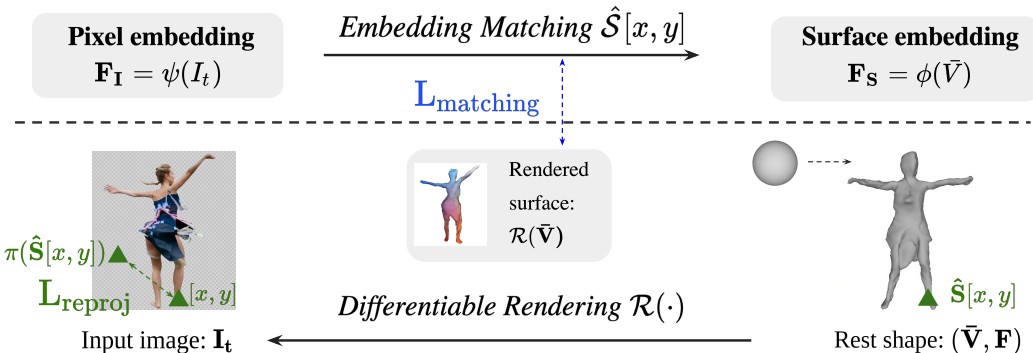

Figure 2: We learn a joint pixel-surface embedding space for dense correspondence between pixels in video frames $I_t$ and points on a canonical 3D surface $(\bar{\mathbf{V}}, \mathbf{F})$. Such embedding space is optimized through "top-down" differentiable rendering $\mathcal{R}(\cdot)$ and "bottom-up" correspondence matching $\hat{\mathbf{S}}[x,y]$ (Sec 3.2). We introduce a 3D matching loss to optimize the embeddings, where the matched surface locations are encouraged to be close to the rendered surface locations. The embedding further enables articulated shape optimization through a 2D-3D-2D cycle reprojection: pixel $[x,y] \rightarrow$ matched surface $\hat{\mathbf{S}}[x,y] \rightarrow$ re-projected pixel $\pi(\hat{\mathbf{S}}[x,y])$ (Sec. 3.3).

**Dense pose and surface mappings.** DensePose [12] directly maps pixels to the 3D surface of a human body model. It requires large amounts of training data with annotated image-to-surface correspondence and is hard to generalize to other categories. Articulation-aware Canonical Surface Mapping (A-CSM) [20] uses geometric cycle consistency for learning to map pixels to corresponding points on a template shape without using keypoint annotations. However, it requires a pre-defined template shape for each category. Continuous Surface Embeddings (CSE) [29] establishes dense correspondences between image pixels and 3D object geometry by predicting an embedding vector of the corresponding vertex in the object mesh for each pixel in a 2D image. While applicable to multiple categories, CSE requires annotations and only applies to categories in the training set. ViSER requires neither a template shape nor annotations to work on categories in the wild.

**Nonrigid shape reconstruction.** One way to accurately reconstruct articulated shapes is to rely on rich sensor data, *e.g.*, multi-view [15] or depth sensors [30], which requires substantial efforts to setup and reconstruct objects in the wild. For monocular videos/images, one popular approach is to adopt strong 3D shape and pose priors [18, 26, 35, 36, 53, 54] but it works well only on limited categories, whose 3D data are easy to collect. To deal with more nonrigid object categories, a recent trend is to learn a category-level 3D shape model from a collection of images or videos with 2D annotations, such as keypoints and object silhouettes [10, 16, 20, 22, 23, 44, 46, 50]. Although they are able to reconstruct more object categories, such as birds and quadruped animals, the reconstruction usually lacks details, and the level of deformation recovered tends to be low.

Category-agnostic methods, such as nonrigid structure from motion (NRSfM) methods [4, 11, 19, 38] reconstruct nonrigid 3D shapes from a set of 2D point trajectories. However, due to the difficulty in obtaining accurate long-range correspondences [37, 41] they do not work well for videos in the wild. A recent work, LASR [49], uses two-frame optical flow to reconstruct articulate shapes from a monocular video with differentiable rendering. Despite the promising results, LASR does not reason about long-range correspondences and can only reliably reconstruct what is visible in a short video. ViSER establishes reliable long-range correspondence that are robust to moderate shape variations and appearance changes. Thus, ViSER can obtain much higher-quality reconstruction by using either a long video or several videos of a category.

## 3  Approach

Fig. 2 provides an overview of our approach, which follows a typical framework of differentiable rendering [16, 25]. Borrowing the notation from LASR [49], we formalize our task as follows. Given a set of video observations including RGB pixel color, segmentation masks, and optical flow estimates $\{I_t, S_t, u_t\}_{t=\{0,...,T\}}$, our goal is to recover a set of shape and motion parameters $\{\mathbf{S}, \mathbf{D}_t\}$ that produce reconstructions $\{\hat{I}_t, \hat{S}_t, \hat{u}_t\}_{t=\{0,...,T\}}$ that match the video observations. We refer to supplementary material for a complete list of notations defined in the paper.

### 3.1 Preliminaries

We represent an object's shape as a triangular mesh $\mathbf{S} = \{\bar{\mathbf{V}}, \mathbf{F}\}$ with canonical vertices $\bar{\mathbf{V}} \in \mathbb{R}^{3 \times N}$ and a fixed topology (edge connectivity) $\mathbf{F} \in \mathbb{R}^{3 \times M}$. To render an object, we displace mesh vertices with motion parameters $\mathbf{D}_t$, apply a perspective projection with camera intrinsics $\mathbf{K}_t$, and rasterize.

We model vertex motion with root body transformations $\mathbf{G}_0$ and object articulations $\{\mathbf{G}_1, \cdots, \mathbf{G_B}\}$ using linear blend skinning (LBS) [20, 21]. LBS constrains vertex motion by linearly blending $B$ rigid "bone" transformations with a skinning weight matrix $\mathbf{W} \in \mathbb{R}^{B \times N}$, transforming the canonical shape into frame $t$ as

$$\mathbf{V}_{i,t} = \mathbf{G_{0,t}} \left( \sum_b \mathbf{W}_{b,i} \mathbf{G}_{b,t} \right) \bar{\mathbf{V}}_i \tag{1}$$

where $i$ is the vertex index, $b$ is the bone index. Similar to LASR, the root body and bone transformations are represented as the outputs of a pose CNN given an input image, $(\mathbf{G_0}, \cdots, \mathbf{G_B}) = \psi_p(I_t)$.

We define a set of surface properties for rendering, including vertex 3D coordinates, textures and features, and rasterize them in a differentiable manner [25]. We denote the differentiable rendering function that renders the property $\mathbf{C}$ defined on a canonical surface to an image as $\mathcal{R}(\mathbf{C}; \mathbf{V}, \mathbf{W}, \mathbf{G})$, which executes the blending skinning function in Eq. (1) and softly blends the surface property based on their depth and barycentric coordinates [25]. For simplicity, we omit the shape, skinning, and motion parameters parameters and write the differentiable rendering function as $\mathcal{R}(\mathbf{C})$. To render optical flow, we rasterize and project vertex coordinates in two consecutive frames and compute their 2D displacements [49]. Such renderings are compared against video observations to compute gradients for updating model parameters.

### 3.2 Video-specific Surface Embedding

**Pixel-surface embeddings.** We learn pixel and surface embeddings that map corresponding pixels in different frames to the same point on a canonical 3D surface. Intuitively, consider a particular region on the canonical surface mesh that is the "nose" of an articulated human. The surface embedding captures a descriptor for the nose, which can then be matched to pixel-level descriptors at each frame.

Given an input image $I_t$, the pixel-wise descriptor embedding is computed by a U-Net [34] encoder:

$$\mathbf{F_I}[x, y, t] = \psi_e(I_t)[x, y] \in \mathbb{R}^{16}, \tag{2}$$

where $[x, y, t]$ are pixel locations at frame $t$. The surface embedding is computed by a position-encoded MLP:

$$\mathbf{F_S}(X, Y, Z) = \phi_e(X, Y, Z) \in \mathbb{R}^{16}, \tag{3}$$

where $\phi_e(\cdot)$ is an MLP defined over 3D points $(X, Y, Z)$ in the canonical space, augmented with Fourier positional encoding [28]. The two embeddings are optimized on test videos such that pixels representing the same surface location in different frames are mapped to the same canonical surface point [20].

**Correspondence via soft-argmax regression.** Given the pixel and surface embeddings, we construct a per-frame cost volume $D(\mathbf{F_I}, \mathbf{F_S})$ of size $H \times W \times N_s$ over pixels and surface points (we randomly sample $N_s = 200$ surface points at each step) by considering their cosine feature distances,

$$D(\mathbf{F_I}, \mathbf{F_S})[x, y, i] = 1 - \cos\big(\mathbf{F_I}[x, y], \mathbf{F_S}(X_i, Y_i, Z_i)\big). \tag{4}$$

Normalizing the cost volume over the surface point dimension yields a softmax "heatmap" over surface points that potentially match to pixel $(x, y)$, as shown in Fig. 3 (Left):

$$\sigma_{(x,y)}[i] = \frac{e^{-D[x,y,i]/\tau}}{\sum_j e^{-D[x,y,j]/\tau}}, \tag{5}$$

where $\tau$ is a temperature scaling parameter that is jointly optimized with the feature embeddings. To output a single surface point for pixel $(x, y)$, we can compute a "soft" argmax [17, 48] by taking the expectation of the softmax distribution over the 3D locations of the points samples,

$$\hat{\mathbf{S}}[x, y] = \sum_i \sigma_{(x,y)}[i](X_i, Y_i, Z_i), \tag{6}$$

where $(X_i, Y_i, Z_i)$ is the i-th sampled surface point and $\sigma_{(x, y)}[i]$ is the matching probability of pixel $(x, y)$ over the sampled points $i \in \{1, 2, \ldots, N_s\}$.

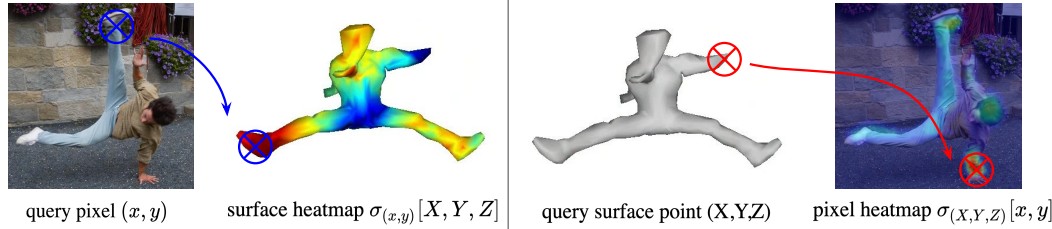

| query pixel $(x,y)$ | surface heatmap $\sigma_{(x,y)}[X,Y,Z]$ | query surface point (X,Y,Z) | pixel heatmap $\sigma_{(X,Y,Z)}[x,y]$ |

Figure 3: Pixel-surface embeddings establish a continuous mapping between pixels and points on a canonical surface. **Left:** Given a query pixel at (x,y), we match it to a set of canonical surface points, where the matching distribution is used to regress a continuous mapping to the canonical surface. **Right:** Given a query surface point (X,Y,Z), a matching distribution over pixels can be computed. Warm color indicates high matching probability.

We can also normalize the $H \times W \times N_s$ cost volume over spatial positions to capture a distribution of pixel locations that match to each surface point $(X_i, Y_i, Z_i)$:

$$\sigma_{(X_i,Y_i,Z_i)}[x,y] = \frac{e^{-D[x,y,i]/\tau}}{\sum_{[x,y]} e^{-D[x,y,j]/\tau}}. \tag{7}$$

and compute a similar soft argmax mapping of surface points to pixels, as shown in Fig. 3 (Right).

**Relation to keypoints.** The output of classic keypoint detectors are often represented as $K$-channel heatmaps over the pixel grid, where $K$ is the number of keypoints. To define dense keypoints, one may increase the number of channels, which is computationally heavy. Similar to CSE [29], we represent dense keypoints as low-dimensional pixel-surface embeddings, which establishes a mapping between pixels and a canonical 3D surface, but far more efficiently. DensePose [12] and CSM [20] use an alternative pixel-to-surface mapping that regresses a surface coordinate at every pixel. In contrast, our pixel-surface embedding captures multimodal uncertainties over keypoints; for example, $\sigma_{(x,y)}[i]$ can capture the fact that a particular pixel matches well to both the left and right ankle, as visualized in Fig. 3, while a regressor may "regress" to the mean of the two surface coordinates.

### 3.3 Learning Embeddings and Articulated Shapes

Next we will introduce the loss functions that enable learning both embeddings and articulated shapes from monocular videos without a pre-defined shape template or annotated correspondence. To learn non-degenerate embeddings and overcome the local optima issue in differertiable renderers, we carefully construct a 3D matching loss and a 2D cycle loss.

**3D match loss.** Arguably, the simplest loss to learn embeddings is to minimize the difference between the rendered surface features and the observed pixel features:

$$L_{\text{feature-consistency}} = \sum_{x,y} \left( 1 - \cos\left( \mathcal{R}(\mathbf{F_S})[x,y], \mathbf{F_I}[x,y] \right) \right), \tag{8}$$

where $\cos(\cdot)$ denotes the inner product between two normalized vectors, and $\mathcal{R}(\mathbf{F_S})$ is the differentiably rendered surface descriptors. However, the feature consistency loss admits a trivial solution, where all pixel and surface features are the same constant (yielding zero error). To address this, we introduce a 3D matching loss that ensures pixel embeddings only match to surface embeddings rendered at the pixel location:

$$L_{\text{matching}} = \sum_{x,y} \left\| \mathcal{R}(\bar{\mathbf{V}})[x,y] - \hat{\mathbf{S}}[x,y] \right\|_2 \tag{9}$$

where $\mathcal{R}(\bar{\mathbf{V}})$ is the rendered 3D surface location and $\hat{\mathbf{S}}[x,y]$ is the estimated pixel-to-surface mapping from Eq. (6), computed through sampling and computing the softmax distributions $\sigma[i]$ over surface points [17]. To minimize the loss, the embeddings of surface points that do not project to $(x,y)$ will be pulled away from the pixel embedding of $(x,y)$ in a contrastive way [13].

**2D cycle loss.** The match loss aims to learn pixel-surface embeddings that are consistent over video frames and discriminative over difference surface locations. However for articulation optimization, the match loss suffers from bad local optima issue similar to other losses based on differentiable rendering [25]. For instance, when the rendering of a body part is outside the ground-truth object silhouette, a gradient update of articulation parameters would likely not incur a lower loss.

To guide articulated 3D shape learning using the learned pixel-surface embeddings, we further define a cycle-based re-projection loss, inspired by prior approaches in 3D model fitting with keypoints [3] and canonical surface mappings [20]. Given an input image, we establish a 2D-3D mapping by extracting a pixel embedding and matching it to surface embedding. Then, we compute the expected surface coordinate $\hat{\mathbf{S}}[x,y]$ at every pixel using Eq. (6), and ensure the differentiably rendered canonical surface coordinate lands back on the original pixel coordinate $(x,y)$,

$$L_{\text{reproj}} = \sum_{x,y} \left\| \mathcal{R}(\hat{\mathbf{S}}[x,y]) - (x,y) \right\|_2 . \tag{10}$$

**Reconstruction loss.** Finally, we make use of reconstruction losses to ensure that generated images, masks, and flows match their estimated counterparts:

$$L_{\text{recon}} = \beta_1 ||\hat{S}_t^i - S_t||_2^2 + \beta_2 ||\hat{I}_t^i - I_t||_2^2 + \beta_3 \sigma_t ||\hat{u}_t^i - u_t||_2 + \beta_4 \text{pdist}(\hat{I}_t, I_t) \tag{11}$$

where $\{\beta_1, \cdots, \beta_4\}$ are weights empirically chosen, $\sigma_t$ is the normalized confidence map for flow measurement, and $\text{pdist}(\cdot, \cdot)$ is the perceptual distance [51] measured by an ImageNet-pretrained AlexNet. The reconstruction losses ensure the match between rendered and observed optical flow, texture and silhouette images.

**Regularization.** To avoid degenerate shapes, we use mesh Laplacian regularization [16, 49] to enforce the recovered shape to be smooth, and as-rigid-as-possible (ARAP) regularization to enforce the deformation to be locally rigid [44]. Different from prior work that only preserves the length of edges after articulation, we encourage both the area and length of faces to be the same after articulation. The area preserving term is defined as

$$L_{\text{ARAP-area}} = \sum_{i=1}^{|E|} \sum_{j \in N_i} \left| \left| \mathbf{E_i^t} \times \mathbf{E_j^t} \right| - \left| \mathbf{E_i^{t+1}} \times \mathbf{E_j^{t+1}} \right| \right| , \tag{12}$$

where $|E|$ is the number of edges and $N_i$ the indices of neighbouring edges.

### 3.4 Representing Surface Properties with MLPs

By extending surface embedding MLPs with additional dimensions, we can model other surface properties including textures and even surface-based geometric deformations. Compared to explicitly defined textures, such continuous implicit representations have the capacity to encode arbitrary amount of details and are empirically easier to optimize.

**Surface appearance.** The appearance of the object is represented by a coordinate-MLP queried at points on the canonical mesh surface. To handle view-dependent appearance (such as shadow and lighting), we further concatenate the Fourier features of the $(X, Y, Z)$ coordinates with a frame appearance code, as the input to the texture MLP,

$$\mathbf{C_{i,t}} = \phi_{tex}(\mathcal{F}(\bar{\mathbf{V}}_i), \omega_t) \in \mathbb{R}^3, \tag{13}$$

where $\bar{\mathbf{V}}_i$ is the i-th canonical mesh vertex, which is passed through a Fourier encoder $\mathcal{F}(\cdot)$ as used in NeRF [28], and concatenated with $\omega_t$, a 64-dimensional frame appearance code associated each image frame $t$, predicted from a ResNet-18, as $\omega_t = \psi_{tex}(I_t)$.

**Instance shape deformation fields.** To deal with videos of multiple instances of the same category, as experiments in Sec. 4.3, we model shape variations across instances by a continuous surface deformation field defined on the canonical surface. Similar to the surface texture, we represent the surface deformation field by a shape MLP,

$$\mathbf{V_{i,k}} = \bar{\mathbf{V}}_\mathbf{i} + \phi_{shape}(\mathcal{F}(\bar{\mathbf{V}}_i), \alpha_k) \in \mathbb{R}^3, \tag{14}$$

where $\mathbf{V_k}$ is the rest shape of instance $k$ and $\alpha_k$ is a video-specific 64-dimensional shape code that is randomly initialized and optimized together with the shape MLP.

## 4  Experiments

We evaluate ViSER in three different scenarios where objects are highly articulating, making it challenging to reconstruct and estimate long-range correspondences. First, we consider long human

Table 1: 2D Keypoint transfer accuracy on athletic videos. Methods with [*] use keypoint annotations to train. Best results are in bold.

| Method | break-1 | break-2 | dance | parkour | ballet-1 | ballet-2 | ballet-3 | Ave. |
|---|---|---|---|---|---|---|---|---|
| [*]DensePose CSE [29] | 56.0 | 13.2 | 77.2 | **85.9** | 45.6 | 49.0 | 64.5 | 55.9 |
| [*]VIBE+SMPLify [18] | 37.1 | 8.2 | 70.4 | 83.8 | **55.4** | 53.0 | **78.8** | 55.2 |
| LASR [49] | 29.1 | 18.1 | 56.6 | 49.8 | 44.5 | 47.4 | 48.6 | 42.0 |
| ViSER (Ours) | **70.5** | **22.5** | **80.7** | 62.9 | 52.7 | **56.1** | 59.9 | **57.9** |

Table 2: 2D Keypoint transfer accuracy on multiple elephant videos. Best results are in bold.

| Method | inner | across |
|---|---|---|
| CSE [29] | 55.7 | 52.2 |
| Flow-VCN [48] | 51.1 | 41.2 |
| LASR [49] | 57.8 | - |
| ViSER (Ours) | **80.4** | **68.9** |

Table 3: 2D Keypoint transfer accuracy on BADJA dataset. Best results are in bold.

| Method | camel | dog | cows | horse | bear | Ave. |
|---|---|---|---|---|---|---|
| CSE [29] | 48.8 | 38.6 | 63.8 | 60.2 | 76.6 | 57.6 |
| Flow-VCN [48] | 47.9 | 25.7 | 60.7 | 14.4 | 63.8 | 42.5 |
| N-NRSfM [38] | 67.8 | 17.9 | 70.0 | 8.7 | 60.2 | 44.9 |
| LASR [49] | **81.9** | 65.8 | **83.7** | 49.3 | 85.1 | 73.2 |
| ViSER (Ours) | 80.1 | **73.8** | 82.9 | **76.3** | **87.3** | **80.1** |

videos with loose clothing and unusual poses. Next, we evaluate on videos of articulated animals for which accurate shape templates are missing. Finally, we analyze a multi-video variant of ViSER that learns a single model from multiple videos of the same category. All scenarios require jointly establishing long-range correspondences and reconstructing articulated 3D shapes at the same time.

**Optimization details** We use the AdamW [27] optimizer with a batch of 4 consecutive image pairs. We reconstruct a long video sequence in an incremental manner similar to classic SfM. First, we use an initial set of around 20 consecutive frames to initialize the shape and pixel surface embeddings. The initial set is selected such that the viewpoint coverage is large enough. Then we gradually add in new frames. When a new frame is added, we first apply the 2D cycle loss $L_{reproj}$ to optimize its articulations, and then jointly optimize all frames with all losses. Empirically, simultaneously optimizing all the video frames produces unstable results of root body poses (or equivalently camera poses).

## 4.1 Athletic Video Reconstruction

**Dataset.** To evaluate ViSER on long-videos, we construct an athletic video dataset that is challenging due to loose clothing and unusual body poses. It consists of four videos from DAVIS [31] and three ballet videos. All videos are segmented and manually annotated with keypoints following the MSCOCO format [24]. We only use keypoint annotations for evaluation purposes.

**Metrics.** Due to the lack of ground-truth 3D data for challenging athletic human videos, we use 2D keypoint transfer as a proxy metric [2, 52]. Given any two frames from a video, the goal is to transfer an annotated 2D keypoint from one frame to another. The accuracy is measured by percentage of correctly transferred keypoints over all T(T-1) pairs of frames in a T-frame video. A transferred keypoint is marked as correct when its distance to the ground-truth annotation is lower than $d_{th} = 0.2\sqrt{|S|}$, where $|S|$ is the area of the ground-truth silhouette [2]. In general, a more accurate reconstruction leads to a higher transfer accuracy.

**Baselines.** To compare with template-based approaches for video human reconstruction, we use VIBE with SMPLify temporal smoothing [18]. To compare with template-free methods, we use LASR [49], which also reconstructs articulated shapes using the same input setting as ours. To transfer keypoints from a reference frame to a target frame, we back-project the annotated keypoint in the reference frame to the canonical surface, and then project the intersected 3D point to the target frame. We also compare against Densepose CSE [29], which produces dense pixel-to-surface correspondences for a given category, but does not produce 3D reconstructions. To transfer keypoints for Densepose CSE, we compute pixelwise surface mappings for both frames and find the best matching w.r.t. geodesic distance on the surface. We further qualitatively compare against a state-of-the-art human reconstruction method, PiFUHD [36] in Fig. 4, which only produces reconstruction, but not correspondence.

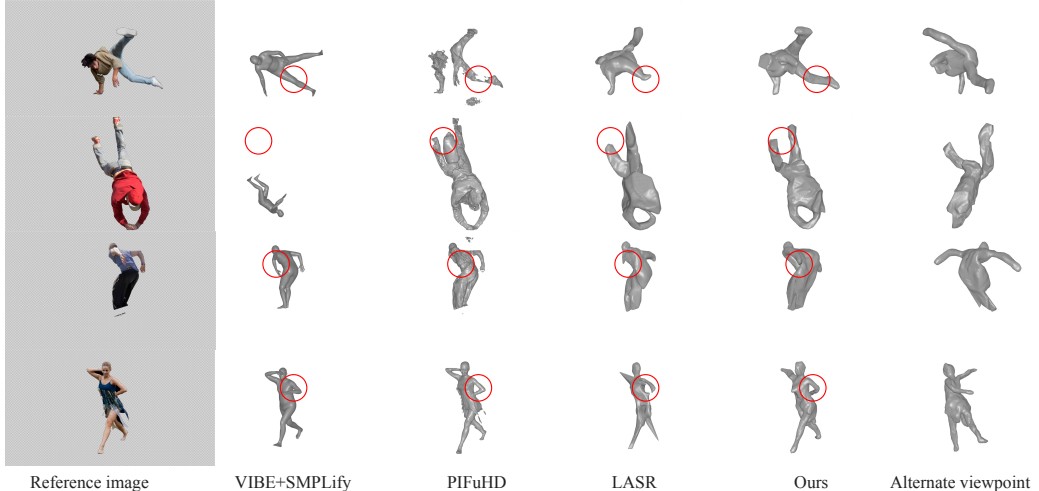

Reference image    VIBE+SMPLify    PIFuHD    LASR    Ours    Alternate viewpoint

Figure 4: Qualitative comparisons for athletic video articulated shape reconstruction. Compared to methods that uses shape and pose priors (VIBE+SMPLify and PiFUHD), our method achieves comparable performance for common appearance and poses, and does much better on unusual poses such as break-dancers.

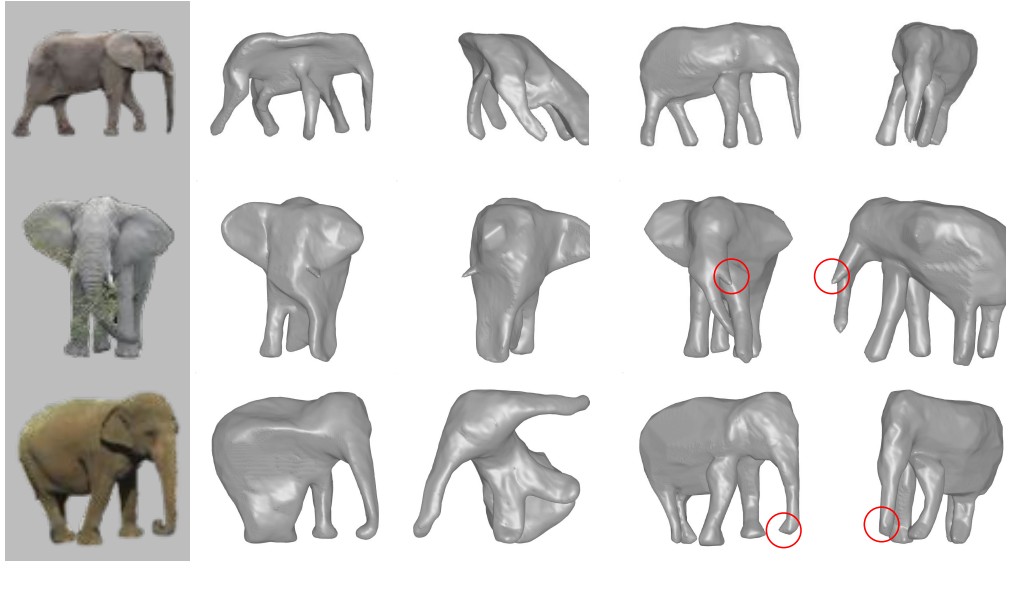

Reference image    LASR    Alternate view    Ours multi-video    Alternate view

Figure 5: Qualitative comparisons for elephant shape reconstruction from multiple videos. Notice that ViSER is able to take advantage of multiple videos to improve the category-level shape reconstruction but also reconstruct instance-specific details (as shown in red circles).

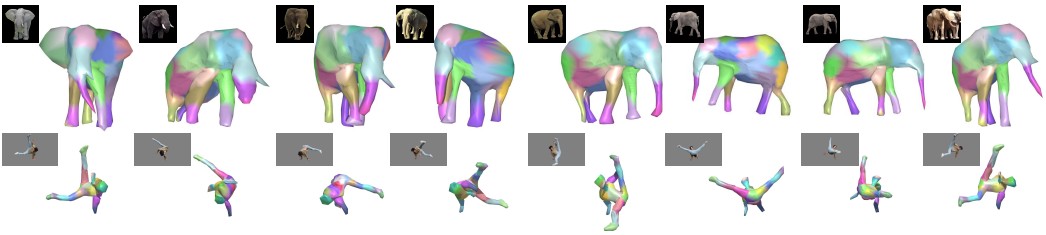

Figure 6: Part segmentation results. Colors are determined by hard-assigning vertices to the closest rigid bones.

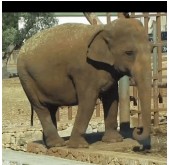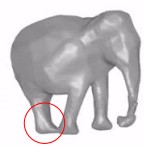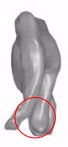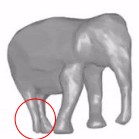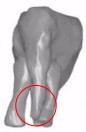

| Reference image | Ours-single video | Ours-multi video |

Figure 7: Comparison between single video ViSER and multi-video ViSER in terms of reconstructing YTVOS elephants. We find using multiple videos helps reconstructing the body parts that may be occluded in a single video. While single-video ViSER reconstructs a flattened shape and misses the hidden rear leg of the elephant, multiple video ViSER reconstructs a more plausible shape and recovers both the two rear limbs.

**Results.** Fig. 4 shows visual reconstruction results on sample videos and for different techniques. ViSER estimates reconstructions that are more faithful to the input than the baselines, especially when the humans have unusual poses like in the first two rows. The accurate long-range correspondence enables ViSER to reconstruct finer details than LASR that does not explicitly try to estimate long-range correspondences. We summarize quantitative comparisons in Tab. 1. There is a moderate performance gap between ViSER and template-based methods when the input fits the latter, such as parkour with tight clothing and usual pose. Note that the supervised Densepose CSE and OpenPose methods fail on breakdance videos due to the novel pose, and also do not work well on ballet dancers due to loose clothing. As a result, template-based approaches that rely on accurate pose recognition, such as VIBE [18] fails. In contrast. our method does not suffer from such poor out-of-distribution generalization. By establishing long-range correspondences, ViSER achieves higher keypoint transfer accuracy and better 3D reconstruction than LASR.

## 4.2 Reconstructing Animals from a Video

We use BADJA [2] to evaluate ViSER on animal videos including camel, cow, dog, bear and horse. Similar to the athletic human video dataset, we compare against template-free methods such as LASR and neural-dense-NRSfM (N-NRSfM) [38]. Similar to LASR and our setup, N-NRSfM learns a video-specific model for object shape, deformation and camera parameters from multi-frame optical flow estimations [8]. We further report performance comparison with dense correspondence methods such as CSE and an optical flow method, VCN. We use the CSE model trained on corresponding animal categories (except that we use the horse model for camel), and the "robust" model of VCN [1], which is the input to our method. As shown in Tab. 3, ViSER achieves better or similar accuracy on all five animal videos compared to LASR and N-NRSfM. While the input optical flow is not robust at estimating long-range correspondences, our method integrates local optical flow to a dense long-range correspondences via a canonical shape, and achieves much better keypoint transfer accuracy. Note that CSE performs well for categories it has been trained on, such as cow, horse and bear, but performs poorly on novel animal categories, such as camel and a novel breed of dog.

## 4.3 Multi-video Shape and Correspondence

We curate a set of seven videos of different elephants from YTVOS [47] for multi-video shape and correspondence recovery. The annotations will be released for further research. We treat multiple videos as a single long video with strong appearance changes and shape variations. In the multi-video setup, We evaluate keypoint transfer accuracy on both the same instance (with video frames) and over different instances (across video frames), as denoted by "inner" and "across". Quantitative results in Tab. 2 shows that ViSER is more accurate than the baseline methods in both cross-video keypoint transfer and inner-video keypoint transfer by a large margin, without using any keypoint annotations or pre-defined shape templates. Fig. 5 show visual result comparisons. While LASR recovers the visible surfaces in a video, it cannot infer the invisible parts. In contrast, our method is able to take advantage of multiple videos from the same category and produce a much better shape reconstruction. Note that LASR cannot handle multiple videos as it requires optical flow computed between every adjacent frame pairs. ViSER, on the other hand, also uses correspondences via estimated 3D shape, thereby allowing the use of multiple videos even when the optical flow is missing across videos.

Table 4: Ablation study on keypoint transfer. Best results are in bold.

| Method | break-1 | elephants-inner | elephants-cross |
|---|---|---|---|
| Full | **70.5** | **80.4** | **68.9** |
| w/o matching loss $L_{matching}$, Eq. (9) | 36.2 | 51.3 | 42.6 |
| w/o reprojection loss $L_{reproj}$, Eq. (10) | 38.3 | 80.1 | 62.5 |
| CSM regression [20] | 47.1 | 77.4 | 63.3 |

**Benefit of Using Multiple Videos.** To examine the benefits of using multiple videos, we further compare multi-video ViSER with single-video ViSER, as shown in Fig. 7. We find using multiple videos helps reconstructing the body parts that may be occluded in a single video.

### 4.4 Part Discovery and Ablations

**Part discovery.** ViSER can discover detailed 3D part segmentation without any manual annotation, as shown in Fig. 6. After training either on a collection of videos or a long video, ViSER can segment the 3D shape into meaningful parts, such as the trunk of the elephants and the feet of the dancer.

**Ablation study.** We perform an ablation study on break-1 and elephants, as shown in Tab. 4. Without the contrastive matching loss, the pixel-surface embedding converges to a trivial solution with a significant decrease of accuracy. Removing the re-projection loss leads to much lower keypoint transfer (KPT) accuracy on the breakdance-1 sequence and cross-video KPT accuracy on the elephant videos. Likely the surface reprojection loss plays an important role in learning correct articulation that follows the bottom-up dense keypoint predictions. This may effectively avoid the local minimum issue for the differentiable rendering optimization. Finally, replacing the pixel-surface embedding with direct CSM regression [20] does not reason about distribution of possible matches and results in worse performance.

**Limitations.** We find ViSER to be sensitive to the random initialization of network parameters. We run optimization with different random seeds for initializing the network parameters and find some perform considerably worse than the others, due to the convergence to bad local optima. Although in practice, one could spot the convergence to a bad local optimum by visualizing the articulated shapes and re-run the optimization with a different random seed, an automatic method for selecting the best model parameters over different trials is desired. We leave how to make the optimization of ViSER robust for future research.

ViSER also relies on optical flow to kick-start with a reasonable initial shape and pose for learning pixel-surface embeddings. Although recent optical flow models generalize well in many scenarios, they may fail when a video is of low resolution or contains significant motion blur. In such challenging cases, using category shape and pose priors to initialize ViSER would be a promising direction.

## 5 Conclusions

We have introduced ViSER, a method to reconstruct articulate shapes, dense trajectories, and object parts from monocular videos. ViSER establishes long-range correspondence by matching 2D pixels to a canonical 3D mesh via learned video-specific surface embeddings. Experimental results show that ViSER, without a template shape or keypoint annotations, compares favorably against prior work on challenging human and animal videos. ViSER shows that it could be fruitful to reconstruct articulate shapes for categories in the wild, and we hope to see more work in this direction.

**Broader impact.** ViSER has many potential applications, *e.g.*, in robotics, AR/VR, and film industry, but may be used for malicious purposes, *e.g.*, producing fake videos or extracting bio-metric information without prior consent. ViSER is only suitable to offline applications as it takes about several hours to process a 80-frame video on one NVIDIA P100 GPU.

## Acknowledgments

This work was supported by Google Cloud Platform (GCP) awards received from Google and the CMU Argo AI Center for Autonomous Vehicle Research. We thank William T. Freeman and many others from CMU and Google for valuable feedback.

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
