# Supplementary Materials for ViSER: Video-Specific Surface Embeddings for Articulated 3D Shape Reconstruction

## A  Appendix

### A.1  Keypoint Transfer Evaluation

We visualize keypoint transfer evaluation procedure in Fig. 1. Keypoints are annotated for meaningful body parts of human and elephants in the MSCOCO format. For human, we annotate fifteen points including "nose, neck, right shoulder, right elbow, right wrist, left shoulder, left elbow, left wrist, mid hip, right hip, right knee, right ankle, left hip, left knee, and left ankle". For elephants, we annotate eleven keypoints including two keypoints for each leg, two keypoint for nose, and one keypoint for tail.

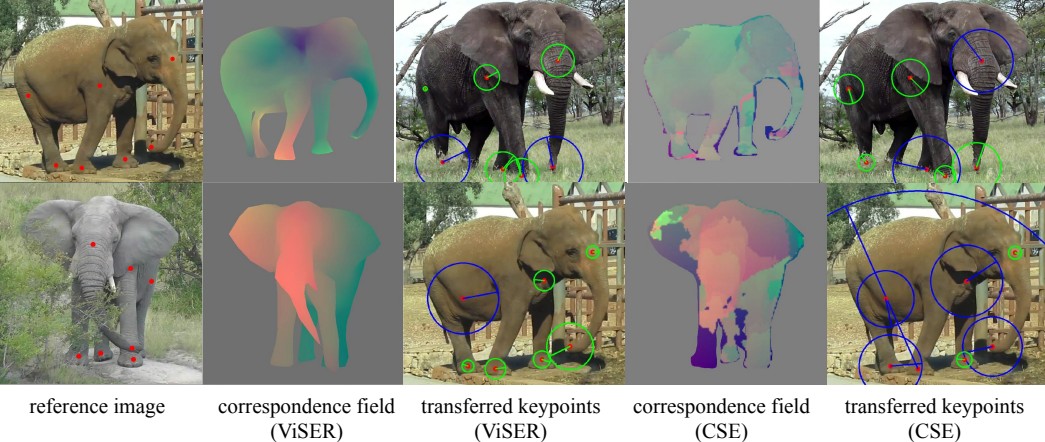

| reference image | correspondence field (ViSER) | transferred keypoints (ViSER) | correspondence field (CSE) | transferred keypoints (CSE) |

Figure 1: Visualization of keypoint transfer results. Red dots indicate annotated keypoints, green circle indicates successful transfer, and blue circle indicates unsuccessful transfer. We transfer keypoints in the reference image with inferred dense correspondence fields as discussed in Sec. 4.1. Such correspondence fields can be established between any two frame in a video, or a set of videos by re-projection via a canonical 3D model. The inferred correspondence field of ViSER is crisper and generally more accurate than CSE [5].

### A.2  Implementation Details

**Surface property rendering** As briefly mentioned in Sec. 3.1, we render surface properties using a differentiable renderer, SoftRas [3]. Specifically, we have

$$\hat{I}_t = \mathcal{R}(\mathbf{C_t}; \mathbf{V_t}, \mathbf{F})  \qquad (1)$$

where $\hat{I}_t$ is the rendered raw pixels, $\mathcal{R}(\cdot)$ is the rasterization function, $\mathbf{C_t}$ is the texture sampled at each mesh vertex position using Eq. (13) of the main text, $\mathbf{V_t}$ is the articulated vertices defined in Eq. (1) of the main text, and $\mathbf{F}$ is the topology of a mesh. We refer readers to LASR [12] for rendering equations of optical flow.

**Surface coordinate-based MLPs** As discussed in Sec. 3.2 and Sec. 3.4, we use coordinate-based MLPs with Fourier encoding to represent canonical surface properties, including surface features $\mathbf{F}$, texture $\mathbf{C_t}$, and instance-specific shape deformation $\mathbf{\Delta V_k}$.

Specifically, we follow NeRF [4] to encode the input $(X, Y, Z)$ coordinates (but not viewing direction) with a set of sine and cosine functions of increasing frequency before passing into an MLP regressor, which is shown useful for learning high-frequency functions from low-dimensional inputs [10].

To encode time-varying texture (due to lighting and shadows), we follow Pixel-NeRF [14] to further learn an image-dependent latent code that modulates the intermediate features of the video-specific texture MLPs. Similarly, to learn instance-specific shape deformation, we learn an instance-specific latent code to modulate the category-specific deformation MLPs.

## A.3  Additional Regularization

**Regularization on bones** Following LASR [13], we use a set of Gaussian "bones" in the canonical space to represent skinning weights. However, we find the bones tend to move outside the surface. We posit that the skinning weights are ambiguous to optimize from limited video observations. In practice, we force the bones to stay around the object surface by minimizing the distance between bone centers and sampled surface points. The distance is measured with Sinkhorn divergence [1], which interpolates between optimal Wasserstein and kernel distances.

**Regularization on surface appearance** To handle time-dependent surface appearances, we use a coordinate-based MLP with an CNN-predicted appearance code (Sec. 3.4). To ensure the appearance does not change drastically over time, we apply temporal $L_1$ regularization on surface appearance,

$$L_{app-reg} = \sum_{i,t} ||\mathbf{C}_{i,t} - \mathbf{C}_{i,t+1}||_1, \tag{2}$$

where $i$ is the index to the mesh vertex and $t$ is the time index.

**Regularization on surface deformation** To handle shape variations over instances of the same category, we use a coordinate-based MLP with a video-specific shape code (Sec. 3.4). To ensure the shape does not change too much over a category, we apply $L_2$ regularization on surface deformation,

$$L_{deform-reg} = \sum_{i,k} ||\mathbf{V}_{i,k} - \bar{\mathbf{V}}_i||_2, \tag{3}$$

where $i$ is the index to the mesh vertex and $k$ is the video index.

## A.4  Comparison to Neural Scene Flow Fields [2]

Related to our problem setup, some recent methods reconstruct dynamic neural radiance fields (NeRF) from a monocular video [2, 6, 7, 11] by differentiable volume rendering, and achieve promising results for novel view synthesis and depth estimation in dynamic scenes.

However, such methods may not work well when the objects exhibit large motion, such as root body rotations. To illustrate this point, we compare with Neural Scene Flow Fields [2] (NSFF), which also use two-frame optical flow as inputs. We ran the public code of NSFF on the DAVIS "dance-twirl" sequence, and the results are shown in Fig. 2.

Specifically, we use COLMAP [8, 9] to estimate camera parameters with regard to the static background (with foreground objects removed). For the "dance-twirl" sequence, COLMAP registers all 90 frames and reconstruct a reasonable background. Then we train NSFF using 4 GPUs for 280k iterations. To extract the reconstructed surface, we sample points from a 256x256x256 grid in the canonical space and run marching cubes. As a result, we find that although NSFF can "overfit" to the input image and optical flow, the extracted surface of the dynamic human is completely off – with streaks connecting the background to the foreground. We hypothesize that one of the possible reasons for NSFF to fail is the lack of long-range correspondence. In contrast, our method is able to deal with root body rotation due to the estimation of long-range correspondences via canonical surface features, as well as the usage of blend-skinning model which regularizes the motion.

Table 1: Table of notations.

| Symbol | Description |
|--------|-------------|
| **Constants** | |
| $T$ | Number of frames in the input video |
| $M$ | Number of faces in the mesh |
| $N$ | Number of vertices in the mesh |
| $B$ | Number of bones for linear blend skinning (LBS) |
| $\mathbf{F}$ | Topology of the mesh |
| $\boldsymbol{\beta}$ | Weights of losses |
| **Input Measurements** | |
| $I_t$ | Input RGB image at time $t$ |
| $S_t$ | Input or measured object silhouette image at time $t$ |
| $u_t^+$ | Input or measured forward optical flow map from time $t$ to $t+1$ |
| $u_t^-$ | Input or measured backward optical flow map from time $t$ to $t-1$ |
| **Renderings** | |
| $\hat{I}_t$ | Rendered color image of the object at time $t$ |
| $\hat{S}_t$ | Rendered object silhouette image at time $t$ |
| $\hat{u}_t^+$ | Rendered forward optical flow map of the object from time $t$ to $t+1$ |
| $\hat{u}_t^-$ | Rendered backward optical flow map of the object from time $t$ to $t-1$ |
| **Shared Model Parameters** | |
| $\bar{\mathbf{V}}_i$ | Position of the i-th mesh vertex of the rest shape |
| $\mathbf{W}$ | Skinning weights matrix |
| $\psi_p(I_t)$ | Weights of the ResNet-18 pose network |
| $\psi_{tex}(I_t)$ | Weights of the ResNet-18 that produces texture code $\omega_t$ |
| $\psi_e(I_t)$ | Pixel embedding, parameterized by a 2D U-Net |
| $\phi_e(X,Y,Z)$ | Surface embedding, parameterized by a coordinate-MLP |
| $\phi_{tex}(X,Y,Z,\omega_t)$ | Surface texture, parameterized by a coordinate-MLP |
| $\tau$ | Temperature parameter for softmax matching distribution over surface points, Eq. (5) |
| **Time-Varying Model Parameters** | |
| $\omega_t$ | A texture code associated with each image $t$ |
| $\mathbf{V_t}$ | Position of mesh vertices at time $t$ |
| $\mathbf{C}_{i,t}$ | Color of mesh vertices at time $t$ |
| $\mathbf{K_t}$ | Intrinsic matrix of a simple pinhole camera (with zero skew and square pixel) at time $t$ |
| $\mathbf{G_{0,t}}$ | Object root body SE(3) transformation at time $t$ |
| $\mathbf{G_{1\ldots B,t}}$ | Bone SE(3) transformations at time $t$ |
| **Additional Parameters for Multi-Video Optimization (Sec. 3.4)** | |
| $\alpha_k$ | A shape code associated with each video $k$ |
| $\phi_{shape}(X,Y,Z,\alpha_k)$ | Video-specific shape deformation from a canonical shape, parameterized by a coordinate-MLP |

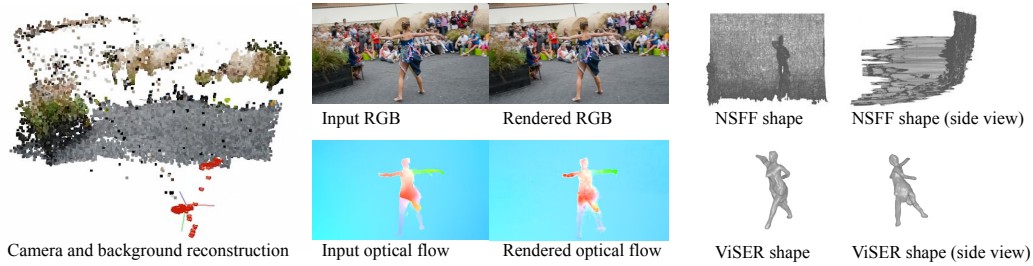

| | | |
|---|---|---|
| | Input RGB | Rendered RGB |
| | NSFF shape | NSFF shape (side view) |
| Camera and background reconstruction | Input optical flow | Rendered optical flow |
| | ViSER shape | ViSER shape (side view) |
| Step 1: Data preparation. | Step 2: NSFF-training. | Step 3: Surface extraction and comparison. |

Figure 2: Comparison to Neural Scene Flow Fields (NSFF) on dance-twirl sequence. Data preparation: We run COLMAP to reconstruct the background and register cameras (in red) of all frames with regard to the reconstructed background. NSFF-training: We train NSFF for 280k iterations on 4 GPUs, and observe that NSFF is able to "overfit" to the input image and optical flow. Surface extraction: we extract surface with marching cubes at a 256x256x256 sampled grid. For NSFF, we observe that the extracted surface of the dynamic human is completely off – with streaks connecting the background to the foreground. In contrast, ViSER is able to correctly reconstruct the dancer.