# OpenReview forum: "ViSER: Video-Specific Surface Embeddings for Articulated 3D Shape Reconstruction"
_NeurIPS.cc/2021/Conference — NeurIPS 2021 Spotlight_

### Official Review · Reviewer_VYGA · 2021-07-08

**Rating:** 6
**Confidence:** 4

**Summary:**

VISER tackles the problem of 3D shape and pose reconstruction from an input video with masks and flows. The main competitor on this task is the recent LASR approach (CVPR 2021) The main technical novelty in VISER is joint embedding of pixels and surface points learned with a self-supervised contrastive loss. These embeddings match each surface points to all corresponding pixels in the video, thus addressing long-range correspondences in the video. ViSER outperforms the state-of-the-art on human and animal videos and the supplementary videos are very impressive.

**Main Review:**

The paper shows impressive results in the supplementary videos and clearly outperforms the state-of-the-art. The main novelty is the joint pixel and surface embeddings, and how it is used to learn long-range correspondences. The main weakness is writing which is hard to follow.

# Writing :

Though the results are impressive, the quality of the writting could be improved. Some parts are still not very clear to me :

-- Please highlight the similarities and differences between VISER and LASR. Is it reasonnable to think that the core of VISER is the feature embeddings i.e. you could take LASR code and data, add a CNN head to predict per pixel features, an MLP head to predict per surface point features, add the embedding losses and improve over LASR? If so, it would greatly help understand how the paper relates to the literature to clarify this.

-- Please clarify the exhaustive list of optimized parameters and their shape as done in LASR.’

-- Please clarify equation (8) and (9) with a figure. In general I find the notation R(F) or R(V) uncommon. Maybe it would help to clarify it a bit more beforehand.

-- The words “top-down” and “bottom-up” are used multiple times and confuse me.

--Equation 13 and 14 are not clear. If V_k is a rest shape, then it belongs to R^(N,3) instead of R^3. What is instance k? Do you mean time-step k?

--L191 : “position-and-image-encoded MLPs”→ I don’t know what you mean, a sentence would be better

-- Please add the losses in the pipeline figure. They are critical to get an intuition on why the method works and brings an improvement over LASR

-- Are the motion parameters G_0 … G_B, K represented as the output of a CNN, like in LASR, or explicit parameters of the optimization problem?


# Losses :

I briefly describe the additional losses over LASr as a sanity-check that I understood the approach.

**consistency** : Each 3d point should have the same features as the pixel it projects to.

**constrastive** : each 3D point should follow this cycle : 3d -> projection to pixel space -> estimated corresponding 3D point via cost volume matching.

**Lreproj** : is the same as the constrastive loss except the cycle is 2d->3d->2d instead of 3d->2d->2d . Why place them in separate paragraphs?

Please clarify what you mean by “Different from geometric cycle consistency loss [19] that suffers from depth ambiguity along the back-projected ray, our formulation leverages graphics rasterization to automatically handle self-occlusion during joint embedding learning.”

# Small comments

**Figure 5:** Some parameters of LASR could be shared across several videos of the same category to compare more fairly with VISER in 4.3 in particular the template shape V and the skinning weights.

**Figure 2 caption :** “Given a <test\> video." There is no training nor testing set in your approach, it is a per-video optimization. Referring to “test” here is confusing.

# Feedback
An additional interesting loss you could consider on the embeddings is enforcing that the distance between two embeddings equals the geodesic distance of the corresponding surface points, as is done in HumanGPS which in their own words “empirically leads to accurate, smooth, and robust results.”

# Post-Rebuttal comment
I have read the other reviews and thank the authors or their responses. I thank the authors for clarifying a number of points that were not clear to me initially. I maintain my initial rating and recommend this paper for acceptance.


**Time Spent Reviewing:**

5 hours

---

> ### Author Response · Authors · 2021-08-10
> **Response to Reviewer VYGA**
>
> Thanks for your constructive feedback. We will incorporate the suggestions on writing in the revised version. Below please find our responses to your questions/comments.
>
> **Q1: Please highlight the similarities and differences between ViSER and LASR.**
>
> The major contribution of ViSER is the models and losses for pixel-surface embeddings. We also remove a few unnecessary components of LASR, summarized as follows:
>
> |Difference | LASR | ViSER-Ours|
> |---|---|---|
> |**Model**|
> | Pixel encoder | N.A | 2D UNet |
> | Surface encoder | N.A | coordinate MLP |
> | Texture | vertex texture | coordinate MLP |
> | # camera hypotheses | 16 | 1 |
> |**Losses**|
> | Consistency loss $L_{cs}$ | No | Yes |
> | Contrastive matching loss  $L_{ct}$ | No | Yes |
> | Re-projection loss $L_{reproj}$ | No | Yes |
> | Symmetry losses | Yes | No |
> | Deformation regularization | ARAP | ARAP-area, Eq. (12) |
> | Weight of the least motion regularization | 1 | 0.01 |
>
>
> **Q2: Please clarify the exhaustive list of optimized parameters and their shape as done in LASR.**
>
> Besides the parameters optimized in LASR, we summarize the additional parameters as follows, and will add a complete list to the revised version.
>
> |Name | Description|
> |---|---|
> | $\psi(I_t)$ | Pixel embedding, parameterized by a 2D U-Net taking image as input |
> | $\phi(X,Y,Z)$ | Surface embedding, parameterized by a coordinate-MLP|
> | $\omega(I_t)$ | A texture code associated with each image, predicted from a ResNet taking image as input |
> | $\phi_{tex}(X,Y,Z,\omega)$ | Surface texture, parameterized by a coordinate-MLP|
> |**Additional parameters for multi-video optimization (Sec 4.3)**|
> | $\alpha_{ \langle 1,\dots,K\rangle }$ | A shape code associated with each video |
> | $\phi_{shape}(X,Y,Z, \alpha)$ | Shape deformation from a canonical shape to each individual object, parameterized by a coordinate-MLP|
>
>
> **Q3: Please clarify equation (8) and (9) with a figure. In general I find the notation R(F) or R(V) uncommon. Maybe it would help to clarify it a bit more beforehand.**
>
> Please see **[here](https://www.dropbox.com/s/64nd43chaklvofa/updated-pipeline.png?dl=0)** for the figure explaining losses defined in Eq. (8) and Eq. (9).
>
> We clarify the definition of $\mathcal{R}(\cdot)$ in Sec 3.1 (Preliminaries) as follows,
>
> "We denote the differentiable rendering function that renders the property ${\bf C}$ defined on a canonical surface to an image as $\mathcal{R}({\bf C}; {\bf V}, {\bf W}, {\bf G})$, which executes the blending skinning function in Eq. (1) and softly blends the surface property based on their depth and barycentric coordinates ("Soft rasterizer: A differentiable renderer for image-based 3d reasoning." CVPR. 2019.). For simplicity, we omit the shape, skinning, and motion parameters parameters $\langle {\bf V}, {\bf W}, {\bf G} \rangle$ and write the differentiable rendering function as $\mathcal{R}(\cdot)$."
>
>
> **Q4: Confusion on the usage of “top-down” and “bottom-up”.**
>
> We use the term “top-down” to describe the differentiable rendering function that maps properties of a canonical model to pixels. The term “bottom-up” refers to the matching from pixels to the canonical surface. We will clarify this in the revised version.
>
> **Q5: What is instance k?**
>
> ${\bf V}_k$ refers to the rest shape of the object in the k-th video, in the multi-video setup (Sec. 4.3).
>
> **Q6: Clarification on “position-and-image-encoded MLPs” and Equation (13)-(14)**
>
> We clarify the description of **surface texture MLPs** as well as Eq. (13) defined in line 191-194 as follows:
>
> The appearance of the object is represented by a coordinate-MLP queried at points on the canonical mesh surface. To handle view-dependent appearance (such as shadow and lighting), we further concatenate the Fourier features of the $(X,Y,Z)$ coordinates together with a appearance code, as the input to the MLP $\phi_{tex}(\cdot)$,
>
> $
> {\bf C_{i,t}}=\phi_{tex}(\mathcal{F}(\bar{\bf V}_{i}), {\omega}_t)\in\mathbb{R}^{3},
> $
>
> where each canonical mesh vertex ${\bf \bar{V_i}}$ is passed through a Fourier encoder $\mathcal{F}(\cdot)$ as used in NeRF, and concatenated with $\omega_t$, a 64-dimensional per-frame appearance code. The appearance code is predicted by a lightweight ResNet taking an image frame as input, as $\omega_{t} = \omega(I_t)$.
>
> We clarify the description of **neural implicit deformation fields** as well as Eq. (14) in line 195-198 as follows:
>
> To deal with videos of multiple instances, we model shape variation across instances by a continuous surface deformation field defined on the canonical surface by a coordinate-MLP. Similar to the surface texture, we concatenate the Fourier features of the $(X,Y,Z)$ coordinates together with a instance shape code, as the input to the MLP $\phi_{shape}(\cdot)$,
>
> ${\bf V_{i,k}}={\bf {\bar V_i}} + {\phi}_{shape}(\mathcal{F}({\bf \bar{V}_i}),\alpha_k)\in\mathbb{R}^{3},$
>
> where ${\bf V_{i,k}}$ is $i$-th vertex of the instance mesh of video $k$ and $\alpha_{k}$ is a randomly initialized 64-dimensional shape code associated with each video.
>
> We will update the descriptions in the revised version.
>
> **Q7: Please add the losses in the pipeline figure.**
>
> The modified pipeline figure and captions can be found **[here](https://www.dropbox.com/s/64nd43chaklvofa/updated-pipeline.png?dl=0)**.
>
> **Captions**: Given a video, we learn a joint pixel-surface embedding space for dense correspondence between pixels in video frames $I_t$ and points on a canonical 3D surface (${\bf \bar{V}, F}$). Such embedding space is optimized through “top-down” differentiable rendering $\mathcal{R}(\cdot)$ and “bottom-up” correspondence matching $\hat{\bf S}[x,y]$ (Sec 3.2). We introduce consistency and contrastive reconstruction losses to optimize the embeddings, where the rendered embedding is encouraged to be similar to the predicted pixel embedding through consistency loss, and the matched surface locations are encouraged to be close to the rendered surface location through contrastive loss. The joint embedding further enables articulated shape optimization through dense surface re-projection following a 2D-3D-2D cycle: pixel $[x,y]\rightarrow $ matched surface ${\bf \hat{S}}[x,y]\rightarrow $ re-projected pixel $\pi({\bf \hat{S}}[x,y])$ (Sec. 3.3).
>
>
> **Q8: Are the motion parameters $\{G_0 \dots G_B\}$, represented as the output of a CNN, like in LASR, or explicit parameters of the optimization problem?**
>
> Same as LASR, the motion parameters $\{G_0 \dots G_B\}$ are represented as the outputs of a 2D CNN.
>
> **Q9: $L_{reproj}$ : is the same as the contrastive loss except the cycle is 2d->3d->2d instead of 3d->2d->3d. Why place them in separate paragraphs?**
>
> Your understanding of the losses is correct. The reason for placing them in different paragraphs is that the two losses serve different and complementary roles.
>
> Specifically, while $L_{ct}$ is useful for learning the embedding space, it suffers from the bad local optimum issue for articulation optimization. While $L_{reproj}$ is useful for articulation optimization, it has multiple zero solutions for embedding matching. Please see more discussion on the two losses in [the reply to reviewer jCtU-Q3](https://openreview.net/forum?id=-JJy-Hw8TFB&noteId=Dr9KTb_G5v).
>
> **Q10: Please clarify what you mean by “Different from geometric cycle consistency loss [19] that suffers from depth ambiguity along the back-projected ray, our formulation leverages graphics rasterization to automatically handle self-occlusion during joint embedding learning.”**
>
> We clarify as follows. The geometric cycle consistency loss function $L_{reproj}$ suffers from the problem of multiple zero solutions for embedding matching. For instance, when the back-projected ray of a pixel at $(x,y)$ intersects at multiple surface locations (a common case is the intersections at the front and back surface), an estimated match ${\bf {\hat S}}[x,y]$ to all intersections will result in zero re-projection loss, which is undesirable. In contrast, $L_{ct}$ uses the graphics rasterization function that renders the 3D coordinates of only the *visible* surface, and compare the estimated matches to the 3D coordinates of the *visible* surfaces. Therefore, $L_{ct}$ does not suffer from this "multiple zero solution problem". We will clarify in the revised version.
>
> **Q11: Some parameters of LASR could be shared across several videos of the same category to compare more fairly with VISER in 4.3. In particular the template shape and the skinning weights.**
>
> We would like to clarify that it is nontrivial for LASR to take advantage of a shared template shape across videos, due to the difficulties in registering reconstructions of different videos to the same canonical space. Some discussion on canonicalization can be found in “C3DPO: Canonical 3D Pose Networks for Non-Rigid Structure From Motion. CVPR 2019.” We tried running LASR on multiple videos concatenated together, but the results were worse than the single video LASR baseline due to mis-aligned body parts.
>
> **Feedback: Geodesic distance on the canonical surface might help regularizing the embedding**
>
> Thank you for the suggestion. We agree that the geodesic distance on the canonical surface could be a good regularizer for the embeddings. However, it is not clear whether geodesic distance would over-regularize the embeddings for visually distinctive but geometrically close-by surfaces, such as eyes and nose. We leave this for future investigation.

---

### Official Review · Reviewer_jCtU · 2021-07-15

**Rating:** 8
**Confidence:** 4

**Summary:**

This paper presents an optimization-based framework that recovers 3D shape, articulated pose, and weak texture from a monocular video of an articulated object. The proposed method extends a recent method (LASR) with a surface embedding matching module for establishing long-range correspondences, and achieves significantly better results on highly deformable objects, such as break dancers.

This is an extremely challenging task, and has a long history in computer vision. Current methods either require extra information, eg depth or keypoints, or rely on category-specific template shape models. The results achieved by this method is very impressive, relying only on off-the-shelf category-agnostic segmentation and optical flow models.

The paper also demonstrates the possibility of extending the method with different instances in multiple videos, which presents exciting potential of capturing category-level priors for fast inference.


**Ethical Concerns:**

I do not see immediate ethical concerns.

**Limitations And Societal Impact:**

A few failure modes are discussed in the supplementary material, but it can be helpful to also show some visualizations of them.


**Main Review:**

## Strengths
### S1 - Method
The proposed extension from LASR is very powerful. It aims at establishing long-range correspondences by matching pixels in different frames through 3D surface embeddings. Specifically, pixels correspondences are established indirectly by matching CNN-predicted pixel embeddings to the same 3D surface embeddings.
At a high level, instead of relying on pixel colors for matching correspondences, it matches correspondences in a learned embedding space, which is more robust against pixel variations resulting from shading, occlusion, deformation etc.
This idea is carefully implemented in this paper and achieves impressive results.


### S2 - Results
- The reconstruction results show significant improvement over the LASR baseline, especially on longer videos, confirming the effectiveness of this embedding correspondence matching module.
- The evaluation and comparison with prior arts are extensive and thorough, presenting lots of technical insights.
- Moreover, the results on multi-video optimization also look very promising, demonstrating the possibility of extracting category-level shape priors from a video collection, which could potentially be extended to unlimited Internet videos at scale.


## Weaknesses
### W1 - Complicated optimization pipeline
One major limitation of the proposed method is its overwhelming complexity. The method is built on top of LASR which already has a very complicated optimization framework, and this new version further adds in even more complexity. It is very impressive that the authors manage to produce those results with this gigantic framework, but it still poses concerns on generalization. However, the paper already shows results on a diversity set of examples, which suggests it's still fairly robust.

### W2 - Texture
Another weakness of the method is the poor texture prediction. I wonder why the optimized texture looks very coarse as shown in the supplementary results.
But as the method mainly focuses on shape reconstruction, and the model bypasses establishing correspondences via pixel colors but rather through embeddings, I think this is not a big issue.


### W3 - Model (more like a question)
Intuitively, I thought the _contrastive matching loss_ $L_\text{ct}$ and the _re-projection loss_ $L_\text{reproj}$ have very similar effects. They are both enforcing the embeddings to be discriminative. The former goes from 2D pixel to 3D surface and compare the rendered 3D coordinate, whereas the latter also goes from 2D pixel to 3D surface, but then back to 2D pixel (also via rendering) and compare the 2D coordinate. But the ablation results in Table 4 suggests using $L_\text{cs}$ alone without $L_\text{reproj}$ also performs much worse than the full model in some cases. Could the authors elaborate on the difference of the two losses?


## Additional comments
- In Fig 5, it would be more convincing for demonstrating the benefit of aggregating multiple videos, if the multi-video results are also compared to single-video results. It would be even more exciting if the authors can show that this learned category prior enables fast adaptation on new videos or even with still images.
- There is a tiny weird formatting issue in the caption of Fig 3.
- It would be helpful to draw a diagram of the three matching losses $L_\text{cs}$, $L_\text{ct}$ and $L_\text{reproj}$.
- Line 273: "rbreak" -> "break"


---
## Post-Rebuttal
I appreciate the authors' effort in the rebuttal, which has addressed all of my questions. The additional results with higher resolution and comparison between single and multiple videos provide further insights. This is a solid submission with a very interesting approach and great results on an extremely challenging task. I remain strongly in favor of acceptance.

**Time Spent Reviewing:**

4.5

---

> ### Author Response · Authors · 2021-08-10
> **Response to Reviewer jCtU**
>
> Thanks for your constructive feedback. We will incorporate the suggestions on writing in the revised version. Below please find our responses to your questions/comments.
>
> **Q1: Does ViSER add more complexity compared with LASR?**
>
> By introducing the pixel-surface embeddings, ViSER actually *removes* a few unnecessary components of LASR:
> * First, we find that the symmetry plane input and symmetry losses are not necessary when a long video (or multiple videos) covering sufficiently different viewpoints is used.
> * Second, we observe the number of camera hypotheses can be reduced from 16 to 1.
> * Lastly, we find that the least motion loss can be reduced by 100 times (although still required to reach a good local optima) to encourage more deformable shapes.
>
> As a result, ViSER is able to deal with more challenging data than LASR, such as large articulations, occlusions and appearance changes. Finally, we summarize the difference between ViSER and LASR in [the reply to Reviewer VYGA-Q1](https://openreview.net/forum?id=-JJy-Hw8TFB&noteId=svoh3ltJMWm).
>
> **Q2: Why are the textures not well-optimized?**
>
> The over-smoothed texture prediction is due to the way we *query* textures during optimization. Specifically, we query texture colors from an MLP at vertex locations (x,y,z) of a relative *low resolution* mesh (~800 vertices), and render them based on depth and barycentric coordinates. Although the texture MLP is capable of representing arbitrarily high-resolution textures, the number of queries is fixed and the locations of the queries do not vary a lot near the end of the optimization.
>
> As shown **[here](https://www.dropbox.com/s/p5z7mhszujvkwss/texture-resolution.png?dl=0)**, if we use a higher-resolution mesh (4k vertices vs 800 vertices) during optimization, a higher quality texture can be recovered with the cost of roughly linearly-increased memory consumption and optimization time.
>
> **Q3: Difference between $L_{ct}$ and $L_{reproj}$**
>
> The two loss functions serve different purposes and are **complementary** to each other:
>
> * In terms of learning a discriminative pixel-to-surface matching embedding, $L_{reproj}$ suffers from the problem of **multiple zero solutions**. When the back-projected ray of a pixel $p$ intersects at multiple surface locations (for instance the front and back surface),  a match of $p$ to all intersections on the surface results in zero loss. In contrast, $L_{ct}$ does not suffer from this issue.
>
> * In terms of optimizing articulation parameters, the loss $L_{ct}$ suffers from the problem of **a flat optimization surface**. For instance, when the rendered silhouette of a body part is outside the ground-truth silhouette, any update of articulation parameters would likely not incur a lower reconstruction loss. In contrast, $L_{reproj}$ provides a better optimization surface, where the change of articulations in the desired direction results in lower re-projection loss.
>
> **Q4: Benefit of using multiple videos**
>
> To examine the benefits of using multiple videos, we further compare multi-video ViSER with single-video ViSER, as shown **[here](https://www.dropbox.com/s/g94bawa0z6oc8se/single-video-viser.mp4?dl=0)**.
>
> **Discussion**: While single-video ViSER reconstructs a flattened shape and misses the hidden rear leg of the elephant, multiple video ViSER reconstructs a more plausible shape and recovers both the two rear limbs. Fast adaptation to new videos/still images is an exciting direction for further work.
>
> **Q5: Diagram of the matching losses**
>
> The modified pipeline figure with the matching losses $\mathrm{L_{cs}}$, $\mathrm{L_{ct}}$, and $\mathrm{L_{reproj}}$ can be found **[here](https://www.dropbox.com/s/64nd43chaklvofa/updated-pipeline.png?dl=0)**.

---

### Official Review · Reviewer_mXo1 · 2021-07-16

**Rating:** 7
**Confidence:** 4

**Summary:**

Authors propose a method for reconstructing deforming objects from videos (with the target object pre-segmented). The output is a mesh with bone assignments, plus temporal deformation parameters (i.e. bone transformations). The approach predicts correspondences between frame pixels and points on the mesh by mapping them into a joint embedding space, using a contrastive loss. The method is demonstrated on videos of humans and animals, where it out-performs a recent baseline.

**Limitations And Societal Impact:**

There is no discussion of limitations. There is brief (but adequate) discussion of broader societal impacts.

**Main Review:**

Method:

++ The proposed approach is novel. More specifically: there is significant commonality with LASR, but I feel there's enough technical novelty in the present work to justify publication.

++ Different parts of the model are clearly motivated and described.

++ The idea of learning a joint embedding space for the 3D surface and the pixels is elegant.

Experiments:

++ The qualitative results on reconstruction are impressive, including on fairly challenging data

++ Quantitative results (though only on keypoint transfer) exceed baselines (which are sensibly-selected modern approaches), apparently by a significant margin

-- There is no discussion of error bars (due e.g. to different initialisations) -- this should be added.

-- There is no direct evaluation of the geometric reconstruction performance. While this is challenging for natural data (lack of gt), it would be valuable to see such results on synthetic data, plus a comparison to baselines.

-- Part-segmentation evaluation is not evaluated numerically -- only qualitative results are shown (which are ok but do not seem particularly accurate). A quantitative evaluation, against a suitable baseline, would be valuable.

-- The predicted textures are never shown.

Misc.:

- Fig.1: typo: "inlcuding"
- 269: typo: "alblations"

---

## Post-rebuttal

The authors' response addresses my concerns sufficiently, and I remain in favor of acceptance having read the other reviews (and respective responses). The sensitivity to initialisation (as reported in the authors' comment below) seems significant, and so this should be discussed in the revised paper. I still think more quantitative evaluation (of geometric reconstruction) would strengthen the paper -- this could be done on synthetic data, or human data with full annotations (e.g. Human3.6M).

**Time Spent Reviewing:**

2

---

> ### Author Response · Authors · 2021-08-10
> **Response to Reviewer mXo1**
>
> Thanks for your constructive feedback. We will incorporate the suggestions on writing in the revised version. Below please find our responses to your questions/comments.
>
> **Q1: Discussion of error bars**
>
> We run optimization with different random seeds for initializing the network parameters. The keypoint transfer accuracy on breakdance-1 and elephants sequences are as follows (Run-1 corresponds to the result of our full method in Tab.4):
>
> |Trial | breakdance-1| elephants-inner | elephants-cross  |
> |:---|:---|:---|:---|
> |Run-1     |70.5 | 80.4  | 68.9 |
> |Run-2     |66.3  |80.2  | 64.1 |
> |Run-3     |60.3  |75.6  | 56.7 |
> |Run-4     |66.6  |78.3  | 61.7 |
> |Run-5     |61.2  |78.6  | 61.2 |
> |Mean (std) | 65.0 (3.8)      | 78.6 (1.7) | 62.5 (4.0) |
>
> **Observation**: We find using different initializations of network parameters results in different final accuracy. For instance, among the five runs on the breakdance-1 video, “run-3” and “run-5” performs considerably worse than the others, due to the convergence to a bad local optima.
>
> **Discussion**: In practice, one could easily spot the convergence to a bad local optimum after a few thousands gradient updates, by visualizing the articulated shapes, and re-run the optimization with a different random seed. One could also automatically select the best set of model parameters by comparing the final reconstruction loss over different trials, which is a good indicator for bad local optimum.
>
> **Q2: Results of predicted textures are not shown.**
>
> We provide videos of the predicted textures for breakdance-1 and elephant-6 **[here](https://www.dropbox.com/s/ywtc1kzeyl97o4e/texture-video.mp4?dl=0)**.
> Please find more qualitative results of predicted textures in the supplementary webpage.
>
> **Q3: Evaluation of geometric reconstruction and part-segmentation**
>
> Thank you for the suggestions. Given the limited time in the rebuttal period, we found it difficult to find synthetic data with high-quality textures and articulations for geometric reconstruction evaluation. We agree those will be valuable to have in the revised version and would greatly appreciate your recommendation of suitable datasets.

---

### Official Review · Reviewer_H2YS · 2021-07-21

**Rating:** 7
**Confidence:** 4

**Summary:**

This paper presents a new method for reconstructing a 3D shape template and dense 3D part trajectories from a monocular video. This work is very similar to a recent work LASR. The main difference, which is also the main contribution, is to establish long-range correspondences by matching the features between each 2D frame to a 3D canonical shape.

**Ethical Concerns:**

No ethical concerns

**Limitations And Societal Impact:**

See above for the limitations. No negative societal impact.

**Main Review:**

This work solves an interesting and important problem, i.e. reconstructing 3D templates and dense trajectories without any 3D coarse geometry or prior knowledge. The idea of establishing correspondences via pixel-to-model registration makes sense, which avoids the drifting issue caused by integrating optical flows frame by frame.
This method is very close to LASR but achieves better quality.
My main concern is that this method lacks novelty. Most of the pipeline components are quite similar to LASR. The main difference is how to establish correspondence. Instead of supervising with frame-to-frame optical flows, this method introduces a shared canonical space for employing the pixel-to-model registration. However, this pixel-to-model matching is widely used in classical computer vision and graphics techniques for registration.
Besides, a comparison with "Neural Scene Flow Fields for Space-Time View Synthesis of Dynamic Scenes" is expected.
Also, the authors are expected to release the code and data.


**Time Spent Reviewing:**

1 hour

---

> ### Author Response · Authors · 2021-08-10
> **Response to Reviewer H2YS**
>
> Thanks for your constructive feedback. Please find our responses to your questions/comments.
>
> **Q1: Is the proposed method for pixel-to-model correspondence novel?**
>
> We agree that the idea of frame-to-model matching is quite general. However, it is still an open problem how to find correspondence between pixels and a canonical model. This is particularly challenging when the shape of the target object is *highly-nonrigid*, and the template shape, as well as the 2D-3D keypoint annotations are *not* provided. ViSER proposes a solution to this challenging problem.
>
> The technical contribution of ViSER is threefold: First, we cast the correspondence problem as the *embedding matching* between pixel descriptors and canonical shape descriptors  (Sec 3.2). Second, we propose *consistency* and *contrastive* reconstruction losses that optimize the embedding space to be consistent over frames and discriminative over different body parts (Sec 3.3). Lastly, we use a dense reprojection loss to *jointly* optimize the articulated 3D shape together with the embedding space.
>
> **Q2: Comparison to “Neural Scene Flow Fields for Space-Time View Synthesis of Dynamic Scenes (CVPR 21)” is expected**
>
> We ran the public code of Neural Scene Flow Fields (NSFF) on the athletic video dataset, and the results can be found **[here](https://www.dropbox.com/s/gltsw0avpcl3oex/nsff-baseline.gif?dl=0)**.
>
> **Setup**: We follow the instructions to use COLMAP to estimate camera parameters given *ground-truth* segmentations. Specifically, for the “dance-twirl” sequence, COLMAP was able to register all 90 frames and reconstruct a reasonable background. Then we train NSFF using 4 gpus for 3 days. To extract the reconstructed surface, we sample points in a 256x256x256 grid in the canonical space and run marching cubes.
>
> **Observation**: Our observation is that although NSFF can “overfit” to the input image and optical flow well, the extracted surface of the dynamic human is completely off -- with streaks connecting the background to the foreground. We hypothesize that the representation of NSFF, which is $(\sigma, {\bf c}) = \mathrm{MLP}(x,y,z,t)$, is not good at modeling dynamic objects with large root body rotations. In contrast, our method is able to deal with root body rotation because of the explicit factorization of root body pose and object articulation.
>
> **Q3: Code and data**
>
> We will make code and data public upon acceptance.

---

> > ### Comment · Reviewer_H2YS · 2021-08-30
> >
> > All my concerns have been addressed. The comparison with NSFF shows that the proposed method significantly outperforms NSFF. Also, the authors promised to release the code. I therefore would like to raise my rating.

---

### Decision · Program_Chairs · 2021-09-27

**Decision:**

Accept (Spotlight)

**Comment:**

This submission received 4 positive final ratings: 7, 7, 8, 6. The reviewers mostly appreciated novelty (while noting high similarity with LASR), clear presentation and strong empirical performance.
The remaining questions and concerns seemed to be addressed in the rebuttal, as acknowledged by the reviewers.
The final recommendation is therefore to accept as a spotlight.